# Genetic associations with temporal shifts in obesity and severe obesity during the obesity epidemic in Norway: A longitudinal population-based cohort (the HUNT Study)

**Maria Brandkvist**[1,2,3]*, **Johan Håkon Bjørngaard**[1,4], **Rønnaug Astri Ødegård**[2,3,5], **Ben Brumpton**[6,7,8], **George Davey Smith**[7,9], **Bjørn Olav Åsvold**[6,10,11], **Erik R. Sund**[4,11,12], **Kirsti Kvaløy**[1,11,12], **Cristen J. Willer**[13], **Gunnhild Åberge Vie**[1,3]

**1** Department of Public Health and Nursing, NTNU, Norwegian University of Science and Technology, Trondheim, Norway, **2** Children's Clinic, St. Olavs Hospital, Trondheim University Hospital, Trondheim, Norway, **3** Obesity Centre, St. Olavs Hospital, Trondheim University Hospital, Trondheim, Norway, **4** Faculty of Nursing and Health Sciences, Nord University, Levanger, Norway, **5** Department of Clinical and Molecular Medicine, NTNU, Norwegian University of Science and Technology, Trondheim, Norway, **6** K.G. Jebsen Center for Genetic Epidemiology, Department of Public Health and Nursing, NTNU, Norwegian University of Science and Technology, Trondheim, Norway, **7** Medical Research Council Integrative Epidemiology Unit, University of Bristol, Bristol, United Kingdom, **8** Clinic of Thoracic and Occupational Medicine, St. Olavs Hospital, Trondheim University Hospital, Trondheim, Norway, **9** Population Health Sciences, Bristol Medical School, University of Bristol, Barley House, Oakfield Grove, Bristol, United Kingdom, **10** Department of Endocrinology, St. Olavs Hospital, Trondheim University Hospital, Trondheim, Norway, **11** HUNT Research Centre, Department of Public Health and Nursing, NTNU, Norwegian University of Science and Technology, Levanger, Norway, **12** Levanger Hospital, Nord-Trøndelag Hospital Trust, Levanger, Norway, **13** Department of Human Genetics, Internal Medicine, and Computational Medicine and Bioinformatics, University of Michigan, Ann Arbor, Michigan, United States of America

* maria.brandkvist@ntnu.no

**Data Availability Statement:** Data from the HUNT Study can be made available to qualified

## Abstract

### Background

Obesity has tripled worldwide since 1975 as environments are becoming more obesogenic. Our study investigates how changes in population weight and obesity over time are associated with genetic predisposition in the context of an obesogenic environment over 6 decades and examines the robustness of the findings using sibling design.

### Methods and findings

A total of 67,110 individuals aged 13–80 years in the Nord-Trøndelag region of Norway participated with repeated standardized body mass index (BMI) measurements from 1966 to 2019 and were genotyped in a longitudinal population-based health study, the Trøndelag Health Study (the HUNT Study). Genotyping required survival to and participation in the HUNT Study in the 1990s or 2000s. Linear mixed models with observations nested within individuals were used to model the association between a genome-wide polygenic score (GPS) for BMI and BMI, while generalized estimating equations were used for obesity (BMI $\geq$ 30 kg/m$^2$) and severe obesity (BMI $\geq$ 35 kg/m$^2$).

researchers upon request to the HUNT Data Access Committee (hunt@medisin.ntnu.no). The data availability policy for the HUNT study is available here: (http://www.ntnu.edu/hunt/data). Data are only available to research groups with a PI affiliated with a Norwegian research institute. The Norwegian Institute of Public Health will consider applications for data from the Tuberculosis screening program through www.helsedata.no. Linkages require ethical clearance by the Regional Ethical Committees, more information is available at https://rekportalen.no/.

**Funding:** Funding: MB was funded by The Liaison Committee for Education, Research and Innovation in Central Norway (https://helse-midt.no/samarbeidsorganet) with project number 90057601 and GÅV was funded by the Norwegian Research Council (https://www.forskningsradet.no/en/) with grant number 250335. JHB was funded by the Norwegian Research Council with grant number 295989. BB and BOÅ work in a research unit funded by Stiftelsen Kristian Gerhard Jebsen (https://stiftkgj.no/what-we-do/k-g-jebsen-centres-of-medical-research/?lang=en); Faculty of Medicine and Health Sciences, NTNU; The Liaison Committee for Education, Research and Innovation in Central Norway; and the Joint Research Committee between St. Olavs Hospital and the Faculty of Medicine and Health Sciences, NTNU; and the Medical Research Council Integrative Epidemiology Unit at the University of Bristol which is supported by the Medical Research Council and the University of Bristol. GDS works in the Medical Research Council Integrative Epidemiology Unit at the University of Bristol (http://www.bristol.ac.uk/integrative-epidemiology/) MC_UU_00011/1. The funders had no role in study design, data collection and analysis, decision to publish, or preparation of the manuscript.

**Competing interests:** I have read the journal's policy and the authors of this manuscript have the following competing interests: GÅV and JHB report grants from The Research Council of Norway during the conduct of the study. The spouse of CJW works at Regeneron Pharmaceuticals. GDS is an Academic Editor on PLOS Medicine's editorial board. MB reports grants from The Liaison Committee for Education, Research and Innovation in Central Norway during the conduct of the study. The authors declare no other competing interests.

**Abbreviations:** BMI, body mass index; CI, confidence interval; GPS, genome-wide polygenic score; HUNT, Trøndelag Health Study; STROBE, Strengthening the Reporting of Observational Studies in Epidemiology.

The increase in the average BMI and prevalence of obesity was steeper among the genetically predisposed. Among 35-year-old men, the prevalence of obesity for the least predisposed tenth increased from 0.9% (95% confidence interval [CI] 0.6% to 1.2%) to 6.5% (95% CI 5.0% to 8.0%), while the most predisposed tenth increased from 14.2% (95% CI 12.6% to 15.7%) to 39.6% (95% CI 36.1% to 43.0%). Equivalently for women of the same age, the prevalence of obesity for the least predisposed tenth increased from 1.1% (95% CI 0.7% to1.5%) to 7.6% (95% CI 6.0% to 9.2%), while the most predisposed tenth increased from 15.4% (95% CI 13.7% to 17.2%) to 42.0% (95% CI 38.7% to 45.4%). Thus, for 35-year-old men and women, respectively, the absolute change in the prevalence of obesity from 1966 to 2019 was 19.8 percentage points (95% CI 16.2 to 23.5, $p < 0.0001$) and 20.0 percentage points (95% CI 16.4 to 23.7, $p < 0.0001$) greater for the most predisposed tenth compared with the least predisposed tenth, defined using the GPS for BMI. The corresponding absolute changes in the prevalence of severe obesity for men and women, respectively, were 8.5 percentage points (95% CI 6.3 to 10.7, $p < 0.0001$) and 12.6 percentage points (95% CI 9.6 to 15.6, $p < 0.0001$) greater for the most predisposed tenth. The greater increase in BMI in genetically predisposed individuals over time was apparent after adjustment for family-level confounding using a sibling design. Key limitations include a slightly lower survival to date of genetic testing for the older cohorts and that we apply a contemporary genetic score to past time periods. Future research should validate our findings using a polygenic risk score constructed from historical data.

## Conclusions

In the context of increasingly obesogenic changes in our environment over 6 decades, our findings reveal a growing inequality in the risk for obesity and severe obesity across GPS tenths. Our results suggest that while obesity is a partially heritable trait, it is still modifiable by environmental factors. While it may be possible to identify those most susceptible to environmental change, who thus have the most to gain from preventive measures, efforts to reverse the obesogenic environment will benefit the whole population and help resolve the obesity epidemic.

## Author summary

### Why was this study done?

- Our genetic propensities for obesity may make it easier for some and more difficult for others to make healthy lifestyle choices, and for those with genetic predisposition to obesity, today's environment may make these healthy lifestyle choices even more difficult.

- Genetic predisposition can be measured with a new genetic tool encompassing over 2.1 million genetic variants associated with BMI.

- How the effects of genetic predisposition to obesity differ as environments are becoming more obesogenic has not been quantified or validated using the genetic tool.

## What did the researchers do and find?

- We assessed the changes in the prevalence of obesity according to genetic predisposition over 6 decades in Norway, with increasing and stabilizing prevalence of obesity, using a genome-wide polygenic score (GPS) for BMI and validating by sibling design.

- Using genetic data from 67,110 individuals aged 13–80 years with repeated height and weight measurements recorded between 1966 and 2019, we found that the prevalence of obesity differed between the participants with the highest and lowest genetic susceptibilities to obesity for all ages at each decade, and the difference increased gradually from the 1960s to the 2000s and then stabilized over the last decade.

- For example, for 35-year-old men and women, the increase in the prevalence of obesity was 20 percentage points greater for the most genetically predisposed tenth compared with the least predisposed tenth.

## What do these findings mean?

- The results indicate that over the past 6 decades, the least genetically predisposed people seem relatively protected from obesity and almost completely protected from severe obesity, whereas the most predisposed people are at risk for both obesity and severe obesity, suggesting that an interplay between genes and an increasingly obesogenic environment could play a role in growing differences in obesity risk between individuals with varying genetic predisposition.

- The findings from this study highlight the need to identify and to address the specific factors that led to the population-wide increase in obesity.

## Introduction

Obesity is recognized as a disease associated with physical and psychiatric multimorbidities [1,2]. Approximately 60% to 80% of adults and 20% to 30% of children in the high-income countries are having overweight or obesity [3,4], while the prevalence in low- or middle-income countries is increasing substantially [5]. For a trait with 40% to 75% cross-sectional heritability [6,7], the body mass index (BMI) is still highly modifiable by the obesogenic environment [8,9]. Obesity can affect everyone regardless of genetic predisposition. However, across all age categories, not only are people with the genetic propensity for obesity at greater risk of excess weight, but also the impact of their genes is greater in the obesogenic environment of recent years [9].

Recently, a genome-wide polygenic score (GPS) was developed as a quantitative measure of inherited susceptibility for obesity [10]. Unlike the genetic risk score based on 97 genetic variants reaching genome-wide significance [11], the GPS encompasses over 2 million common genetic variants associated with BMI. Although not deterministic, this powerful polygenic tool explains 9% of variation for BMI suggesting a 13-kg weight gradient across polygenic score tenths among today's middle-aged adults [10]. To put in context, the change in the mean BMI seen in many countries of the world between 1975 and 2014 is as large or larger than the difference in the BMI from the top to bottom decile of the GPS [12].

Our previous longitudinal analysis of the Trøndelag Health Study (the HUNT Study) [9] provides convincing evidence of the interplay between genes and the environment. This study focused on population BMI and applied a genetic risk score for BMI with 97 gene variants [11] over 5 decades. Utilizing the dramatic changes in our environment from 1966 to 2019, we now apply the more powerful GPS to show that the same trends exist with the prevalence of obesity.

Arguably, the increased disparity in weight between the genetically predisposed and non-predisposed in recent years could be attributed to assortative mating rather than a function of the obesogenic environment [13]. It is logical to assume that children of couples with obesity are likely to inherit a higher genetic risk for obesity and that genetic variance would amplify for each generation, in turn contributing to increasing obesity prevalence [13]. Confounding could also arise from population stratification, when allele frequencies differ between subpopulations, or from dynastic effects, when parental genes influence offspring outcome through other pathways than shared genes [14–16]. As the GPS distribution between siblings sharing a mother and father is random, we use sibling design to compare BMI within families. This is important as it mitigates bias from assortative mating, population stratification, and dynastic effects [14]. Hence, the goal of this study is to quantify and validate the interplay between our genes and the environment.

## Methods

Our study includes 67,110 individuals of European descent aged 13 to 80 years. The study population consists of participants from the HUNT Study (1984 to 2019) linked to previous height and weight measurements in the tuberculosis screening program (1966 to 1969). The entire adult population in the Nord-Trøndelag region was invited to participate in the HUNT Study conducted in 4 waves: HUNT1 (1984 to 1986), HUNT2 (1995 to 1997), HUNT3 (2006 to 2008), and HUNT4 (2017 to 2019). The Young-HUNT Study, recruiting all teenagers aged 13 to 19 years in the Nord-Trøndelag region, was conducted in 1995 to 1997, 2000 to 2001, 2006 to 2008, and 2017 to 2019. Despite participation decline from 88% in HUNT1 to 70% in HUNT2 and subsequently 54% in HUNT3 and HUNT4, the HUNT Study is considered as the representative of the Norwegian population [17]. The tuberculosis screening program was established in 1943 and contributed to the surveillance of tuberculosis in the general Norwegian population [18]. From the tuberculosis screening data, we limited height and weight data to the time interval with most observations from 1966 to 1969 and excluded participants younger than 14 years as they were not targets for total population surveillance.

### BMI assessment

Measurements were standardized with weight measured to the nearest half kilogram with the participants wearing light clothes and no shoes, and height was measured to the nearest centimeter [19]. BMI was calculated using the formula weight in kilograms per meter squared. As defined by the World Health Organization, we refer to overweight as having BMI greater than or equal to 25 and to obesity as having BMI greater than or equal to 30 [20]. We chose to refer to severe obesity as having BMI greater than or equal to 35. As previously described, we calculated BMI z-scores for participants younger than 18 years [9]. By definition, BMI encompasses adjustments for height. However, a BMI of 30 does not necessarily have the same significance for a tall person as for a short person [21]. In the statistical analyses, we adjusted BMI for height to account for any effect of the 6-cm height increase in the population since the 1960s [22]. As suggested by the reviewers, we repeated the analyses using BMI without adjustment for height and compared the results.

## Genotyping and computation of genome-wide polygenic score (GPS)

Genetic analyses were performed on blood samples collected from adults participating in HUNT2 and HUNT3 [23]. Genotyping was carried out with 1 of 3 different Illumina Human-CoreExome arrays (HumanCoreExome12 v1.0, HumanCoreExome12 v1.1, and UM HUNT Biobank v1.0, Illumina, California, United States of America), as described previously [9,24]. Imputation was performed using minimac3 (https://imputationserver.sph.umich.edu/) from a panel combined from the Haplotype Reference Consortium and 2,202 HUNT low-pass sequenced individuals with indel calling. We constructed a GPS using weights from the polygenic score for BMI derived and validated by Khera and colleagues. Detailed information on the polygenic score derivation and validation is described previously [10]. The GPS of Khera and colleagues includes 2.1 million common variants previously identified to be associated with BMI [11,25]. Palindromic polymorphisms were excluded; however, all available variants of sufficient quality were included regardless of the *p*-value of their association with BMI. Using a Bayesian approach, a posterior mean effect size was calculated for each variant encompassing the extent at which similarly associated variants correlated with each other in a reference population [10]. We included 2.07 million of the 2.1 million common variants, excluding those with insufficient quality of genotyping or imputation in HUNT ($r^2 < 0.8$).

## Statistical analysis

Prior to commencing the analyses, we modified the prospective analysis plan slightly in agreement with all coauthors. The prospective protocol and reasoning for modifications are included in the supplementary files (S1 Protocol, S1 Text). BMI was originally planned as a secondary outcome, and severe obesity was added as an outcome. As suggested by the editor, we present the distribution of participants according to both the GPS and the previously used GRS (Table A in S2 Text). We also show the mean BMI in the bottom and top tenths of genetic predisposition according to each score (Table B in S2 Text). These tables highlight the differences between the 2 genetic scores.

We assessed the association between GPS and BMI using linear multilevel models with observations nested within individuals. To assess linearity, we modeled the association between the GPS and BMI using linear splines with 9 knots according to percentiles of the distribution. We adjusted for sex and time of measurement as categorical variables and used linear splines with knots at every 20 years to adjust for age. We also adjusted for 20 principal components and genotyping batch. Further, we allowed the effect of the GPS to differ according to time of measurement, sex, and age using interaction terms for each. Although we adjusted for age using splines, we used 20-year age categories for the interaction terms. The association between the GPS and BMI was fairly linear justifying a linearity assumption for GPS (Fig A in S2 Text). Hence, for the main analyses, we divided the study population into 10 equally sized groups, the top tenth being the most genetically susceptible to higher BMI and the bottom tenth being the least genetically susceptible. We then estimated the effect of genetic risk of obesity on height-adjusted BMI according to time of measurement, age, and sex. In addition to the previously described interaction terms, we included an interaction term between age and time of measurement.

We modeled the association of GPS with obesity and severe obesity using generalized estimating equations. We included the same covariates as in the models assessing height-adjusted BMI. In the main text, we present results for adults aged 25 to 55 years, as this age band shows a relevant age span and was most complete in our dataset. Based on these models, we plotted the estimated height-adjusted BMI and the prevalence of obesity and severe obesity for the highest compared with the lowest tenth of genetic susceptibility to BMI for chosen ages at each

decade for men and women. We report *p*-values from regression models based on Wald tests and marginal differences based on the delta method, both 2-sided and with significance level at 0.05.

Since the transmission of alleles from parent to offspring is random, the siblings have equal likelihood of inheriting any given genetic variant. To assess whether assortative mating, dynastic effects, or population stratification influenced the results, we analyzed the association of the GPS with height-adjusted BMI as well as with the prevalence of obesity within and between siblings. The sibships' GPS average and each sibling's deviation from the group GPS average were calculated and included as independent variables in the regression models with individuals clustered in sibships. We used maximum likelihood random-effects regression models for height-adjusted BMI and generalized estimated equation models for obesity and severe obesity. We adjusted for age with splines as previously described and included interaction terms between sex and each of the aforementioned measures of genetic predisposition. The within-sibship coefficient is an estimate for differentially genetically exposed siblings. Between-sibship coefficients exceeding the within-sibship coefficients would indicate confounding at the sibship level. Unlike the main analyses, we performed these models separately by time point with 1 observation per individual, assuming the association of GPS with BMI and with obesity to be linear and constant over different ages.

To assess the possibility of selection bias, we estimated the association between obesity status in the 1960s and availability of genetic data. We compared the estimated BMI and the prevalence of obesity among 38,378 individuals excluded due to the lack of genetic data to the estimated BMI and the prevalence of obesity for individuals in our study sample. We used genetic data from first-degree relatives to evaluate if exclusions due to missing genetic data biased the results. To approximate the relative rather than the absolute difference in BMI, we assessed the association between the GPS and the natural logarithm of BMI. Analyses were performed with StataMP 15 (Stata), StataMP 16 (Stata), and Plink 2.0 (cog-genomics.org). This study is reported as per the Strengthening the Reporting of Observational Studies in Epidemiology (STROBE) guideline (S1 Checklist).

## Ethics statement

The study obtained ethics approval from the Regional Committees for Medical and Health Research Ethics with ID 2016/537. All adult participants gave written informed consent before taking part in the HUNT Study. This consent included linkage to other registries. Parents gave written informed consent for adolescents taking part in the Young-HUNT Study.

## Patient and public involvement

There was no patient and public involvement in planning the research question, the outcome measures, the design, or the implementation of the study. Neither were the patients and public asked to assess the burden of participation as only previously collected data were used. Involvement of the Norwegian organization Landsforening for Overvektige will be sought in setting an appropriate method of dissemination.

## Results

The study sample consists of 67,110 participants aged 13 to 80 years with a total of 202,030 BMI measurements, with an average of 3 measurements per person (Fig 1). Although all ages were included in the analyses, we present age specific results for ages 25 to 55 years in the main results. The wider age range is presented in the Supporting information (Figs B and C in S2 Text). In the results, however, we chose to focus on BMI measurements from adult

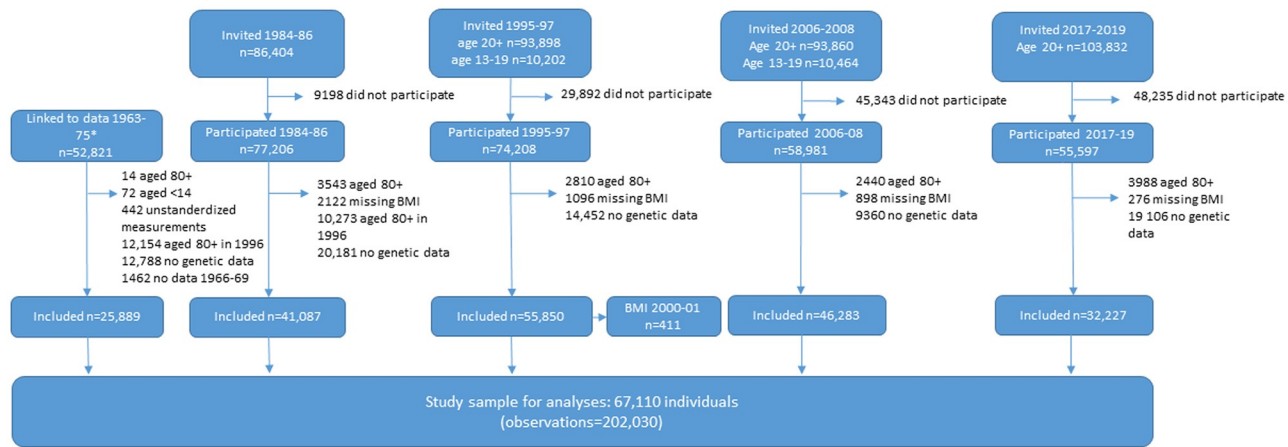

**Fig 1. Flowchart of study participants and criteria for inclusion in study sample.** *Linkage to data from tuberculosis screening program (1963–1975) required participation in HUNT2 (1995–1997) or HUNT3 (2006–2008). Participants could contribute with more than 1 observation in 1966–1969 or in rare cases in 1995–1997 or 2006–2008 by participating in both HUNT and YH. BMI, body mass index; HUNT, Trøndelag Health Study; YH, Young-HUNT.

participants, as observations in the adolescent age range were limited (Table 1). The average age of participants increased gradually from 30 years in the 1960s to 60 years in 2017 to 2019, except for 2000 to 2001 when only adolescents participated. We found an increasing BMI variance and a shift toward a higher prevalence of obesity over time (Table 1, Figs D–F in S2 Text). In the contemporary HUNT population, the GPS explained 8.26% of variance in BMI.

From relative stability in the 1960s to 1980s, the BMI for both the genetically predisposed and non-predisposed increased dramatically from the mid-1980s to the 2000s and then stabilized to a higher level over the past decade. At different ages and decades, estimated height-adjusted BMI differed by 2.5 to 5 BMI units from top to bottom polygenic score tenths (Table 2). While the population BMI increased for both genetically predisposed and lesser predisposed people over time (Table D in S2 Text), the BMI increased more for the genetically predisposed (Table 3, Fig G in S2 Text). For instance, the difference in mean height-adjusted BMI between the most and least genetically predisposed tenths was 1.45 kg/m$^2$ (1.09 to 1.81 kg/m$^2$, $p < 0.0001$) greater in recent years compared with the 1960s. Additional analyses with BMI in models not adjusted for height asserted similar results (Fig H in S2 Text, Tables C, E, and F in S2 Text). We found comparable associations between polygenic risk score and BMI as well as obesity within and between sibling groups with little evidence of bias from assortative mating, population stratification, or dynastic effects (Fig 2, Fig I in S2 Text, Tables G and H in S2 Text). HUNT participants excluded due to missing genetic data had only a slightly higher BMI compared to participants with genetic data [9]. Using genetic data from first-degree relatives, we found no evidence that exclusion due to missing genetic data biased results (Figs J–L in S2 Text). Using the natural logarithm of BMI as the outcome, we found the expected larger effect sizes in more recent years (Table I in S2 Text).

The increase in the prevalence of obesity and severe obesity was observed to be steeper among the genetically predisposed over the time period (Figs 3 and 4). Among 35-year-old men, the prevalence of obesity for the least predisposed tenth increased from 0.9% (95% confidence interval [CI] 0.6% to 1.2%) to 6.5% (95% CI 5.0% to 8.0%, $p$ for difference < 0.001), while the most predisposed tenth increased from 14.2% (95% CI 12.6% to 15.7%) to 39.6% (95% CI 36.1% to 43.0%, $p$ for difference < 0.001). The absolute change in the prevalence of obesity was 19.8 percentage points (95% CI 16.2 to 23.5 percentage points, $p < 0.0001$) greater

**Table 1. Descriptive statistics of male and female participants at each time point (SD and BMI).**

| Year | TBC (1966–1969) | HUNT1 (1984–1986) | HUNT2 (1995–1996) | YH2 (2000–2001) | HUNT3 (2006–2008) | HUNT4 (2017–2019) | Total |
|---|---|---|---|---|---|---|---|
| Men | | | | | | | |
| No. of participants | 12,046 | 19,588 | 26,323 | 155 | 21,187 | 14,487 | 31,717 |
| No. of observations | 12,149 | 19,588 | 26,336 | 155 | 21,192 | 14,487 | 93,907 |
| Mean age (SD) | 30.4 (11.1) | 42.4 (12.6) | 47.6 (16.1) | 18.1 (0.6) | 52.3 (14.5) | 60.2 (11.5) | 47.2 (16.3) |
| Mean BMI (SD) | 23.8 (2.8) | 25.1 (3.0) | 26.3 (3.5) | 22.7 (3.3) | 27.5 (3.8) | 27.9 (4.0) | 26.3 (3.7) |
| Smoking (%) | | | | | | | |
| Never smokers | | | 37.4 | 52.3 | 40.2 | | |
| Former smokers | | | 31.6 | 2.6 | 34.4 | | |
| Current smokers | | | 28.2 | 9.0 | 14.9 | | |
| Missing | | | 2.9 | 36.1 | 8.1 | | |
| Education (%) | | | | | | | |
| Primary | 43.2 | 33.7 | 28.1 | | 19.5 | 13.2 | |
| Secondary | 50.3 | 53.9 | 54.6 | | 58.2 | 60.7 | |
| Tertiary | 5.5 | 12.1 | 15.2 | | 22.0 | 26.0 | |
| Missing | 1.0 | 0.3 | 2.1 | | 0.2 | 0.1 | |
| Chronic disease (%) | | | 31.3 | | 19.3 | | |
| Missing | | | 5.6 | | 29.9 | | |
| Women | | | | | | | |
| No. of participants | 13,843 | 21,499 | 29,527 | 256 | 25,096 | 17,740 | 35,393 |
| No. of observations | 13,975 | 21,499 | 29,545 | 256 | 25,108 | 17,740 | 108,123 |
| Mean age (SD) | 31.0 (11.1) | 42.6 (12.8) | 47.1 (16.5) | 18.2 (0.7) | 51.2 (15.0) | 58.9 (11.9) | 47.0 (16.3) |
| Mean BMI (SD) | 24.1 (3.8) | 24.5 (4.1) | 26.1 (4.6) | 22.8 (3.3) | 26.9 (4.9) | 27.2 (4.9) | 25.9 (4.7) |
| Smoking (%) | | | | | | | |
| Never smokers | | | 45.4 | 46.9 | 42.0 | | |
| Former smokers | | | 21.6 | 2.7 | 29.3 | | |
| Current smokers | | | 29.4 | 17.2 | 19.6 | | |
| Missing | | | 3.6 | 33.2 | 6.5 | | |
| Education (%) | | | | | | | |
| Primary | 50.8 | 41.5 | 33.8 | | 22.2 | 14.6 | |
| Secondary | 43.5 | 47.1 | 46.9 | | 48.7 | 48.1 | |
| Tertiary | 5.0 | 11.1 | 16.8 | | 28.8 | 37.1 | |
| Missing | 0.8 | 0.3 | 2.4 | | 0.3 | 0.1 | |
| Chronic disease (%) | | | 30.1 | | 18.9 | | |
| Missing | | | 7.7 | | 28.9 | | |

Data on smoking and chronic diseases were not available in TBC, HUNT1, and HUNT4 for this project. Data on education were collected from Statistics Norway.

BMI, body mass index; HUNT, Trøndelag Health Study; SD, standard deviation; TBC, Tuberculosis Screening Program; YH, Young-HUNT.

for the highly predisposed. Equivalently for women of the same age, the prevalence of obesity for the least predisposed tenth increased from 1.1% (95% CI 0.7% to 1.5%) to 7.6% (95% CI 6.0% to 9.2%, $p$ for difference $< 0.001$), while the most predisposed tenth increased from 15.4% (95% CI 13.7% to 17.2%) to 42.0% (95% CI 38.7% to 45.4%, $p$ for difference $< 0.001$). The absolute change in the prevalence of obesity among women was 20.0 percentage points

**Table 2. Estimated difference in height-adjusted BMI between the tenths with highest and lowest genetic susceptibilities at various time points.**

| Time range | Age | Men | | | | | Women | | | | |
| | | Difference in BMI | 95% CI | | | p-value | Difference in BMI | 95% CI | | | p-value |
|---|---|---|---|---|---|---|---|---|---|---|---|
| 1966–1969 | 25 | 2.66 | 2.45 | to | 2.86 | <0.001 | 3.53 | 3.33 | to | 3.73 | <0.001 |
| | 35 | 2.96 | 2.74 | to | 3.18 | <0.001 | 3.83 | 3.62 | to | 4.05 | <0.001 |
| | 45 | 3.04 | 2.78 | to | 3.30 | <0.001 | 3.91 | 3.65 | to | 4.18 | <0.001 |
| 1984–1986 | 25 | 2.91 | 2.71 | to | 3.10 | <0.001 | 3.78 | 3.59 | to | 3.97 | <0.001 |
| | 35 | 3.21 | 3.03 | to | 3.39 | <0.001 | 4.08 | 3.91 | to | 4.25 | <0.001 |
| | 45 | 3.29 | 3.09 | to | 3.49 | <0.001 | 4.16 | 3.97 | to | 4.36 | <0.001 |
| | 55 | 3.15 | 2.90 | to | 3.40 | <0.001 | 4.02 | 3.78 | to | 4.27 | <0.001 |
| | 65 | 2.94 | 2.62 | to | 3.25 | <0.001 | 3.81 | 3.49 | to | 4.13 | <0.001 |
| 1995–1997 | 25 | 3.48 | 3.26 | to | 3.70 | <0.001 | 4.35 | 4.14 | to | 4.56 | <0.001 |
| | 35 | 3.78 | 3.60 | to | 3.96 | <0.001 | 4.65 | 4.48 | to | 4.83 | <0.001 |
| | 45 | 3.86 | 3.68 | to | 4.04 | <0.001 | 4.74 | 4.56 | to | 4.91 | <0.001 |
| | 55 | 3.72 | 3.51 | to | 3.93 | <0.001 | 4.60 | 4.39 | to | 4.80 | <0.001 |
| | 65 | 3.51 | 3.24 | to | 3.77 | <0.001 | 4.38 | 4.12 | to | 4.65 | <0.001 |
| 2006–2008 | 25 | 3.98 | 3.71 | to | 4.24 | <0.001 | 4.85 | 4.59 | to | 5.11 | <0.001 |
| | 35 | 4.28 | 4.05 | to | 4.50 | <0.001 | 5.15 | 4.94 | to | 5.37 | <0.001 |
| | 45 | 4.36 | 4.16 | to | 4.56 | <0.001 | 5.23 | 5.04 | to | 5.42 | <0.001 |
| | 55 | 4.22 | 4.02 | to | 4.42 | <0.001 | 5.09 | 4.90 | to | 5.28 | <0.001 |
| | 65 | 4.01 | 3.78 | to | 4.23 | <0.001 | 4.88 | 4.66 | to | 5.10 | <0.001 |
| 2017–2019 | 25 | 4.11 | 3.76 | to | 4.45 | <0.001 | 4.98 | 4.65 | to | 5.31 | <0.001 |
| | 35 | 4.41 | 4.11 | to | 4.70 | <0.001 | 5.28 | 4.99 | to | 5.57 | <0.001 |
| | 45 | 4.49 | 4.23 | to | 4.75 | <0.001 | 5.36 | 5.11 | to | 5.61 | <0.001 |
| | 55 | 4.35 | 4.12 | to | 4.58 | <0.001 | 5.22 | 5.00 | to | 5.44 | <0.001 |
| | 65 | 4.14 | 3.92 | to | 4.35 | <0.001 | 5.01 | 4.80 | to | 5.22 | <0.001 |

BMI, body mass index; CI, confidence interval.

(95% CI 16.4 to 23.7 percentage points, $p < 0.0001$) greater for the highly predisposed (Fig 3, Tables 4 and 5). A similar trend is evident for severe obesity (Fig 4, Tables 4 and 5); the corresponding absolute changes in the prevalence of severe obesity for men and women, respectively, were 8.5 percentage points (95% CI 6.3 to 10.7 percentage points, $p < 0.0001$) and 12.6 percentage points (95% CI 9.6 to 15.6 percentage points, $p < 0.0001$) greater for the highly predisposed. Similar yet slightly smaller changes were found among other ages. (Figs 3 and 4, Tables 4 and 5). With a contemporary prevalence of severe obesity below 2% for most age groups, the least genetically predisposed people seem relatively protected against severe obesity.

**Table 3. Estimated difference in height-adjusted BMI between the tenths with the highest and lowest genetic susceptibilities over time for all ages for men and women combined.**

| Time range | BMI difference | 95% CI | | | p-value |
|---|---|---|---|---|---|
| 1966–1986 | 0.25 | 0.09 | to | 0.41 | 0.002 |
| 1966–1997 | 0.82 | 0.61 | to | 1.03 | <0.001 |
| 1966–2008 | 1.32 | 1.04 | to | 1.59 | <0.001 |
| 1966–2019 | 1.45 | 1.09 | to | 1.81 | <0.001 |

BMI, body mass index; CI, confidence interval.

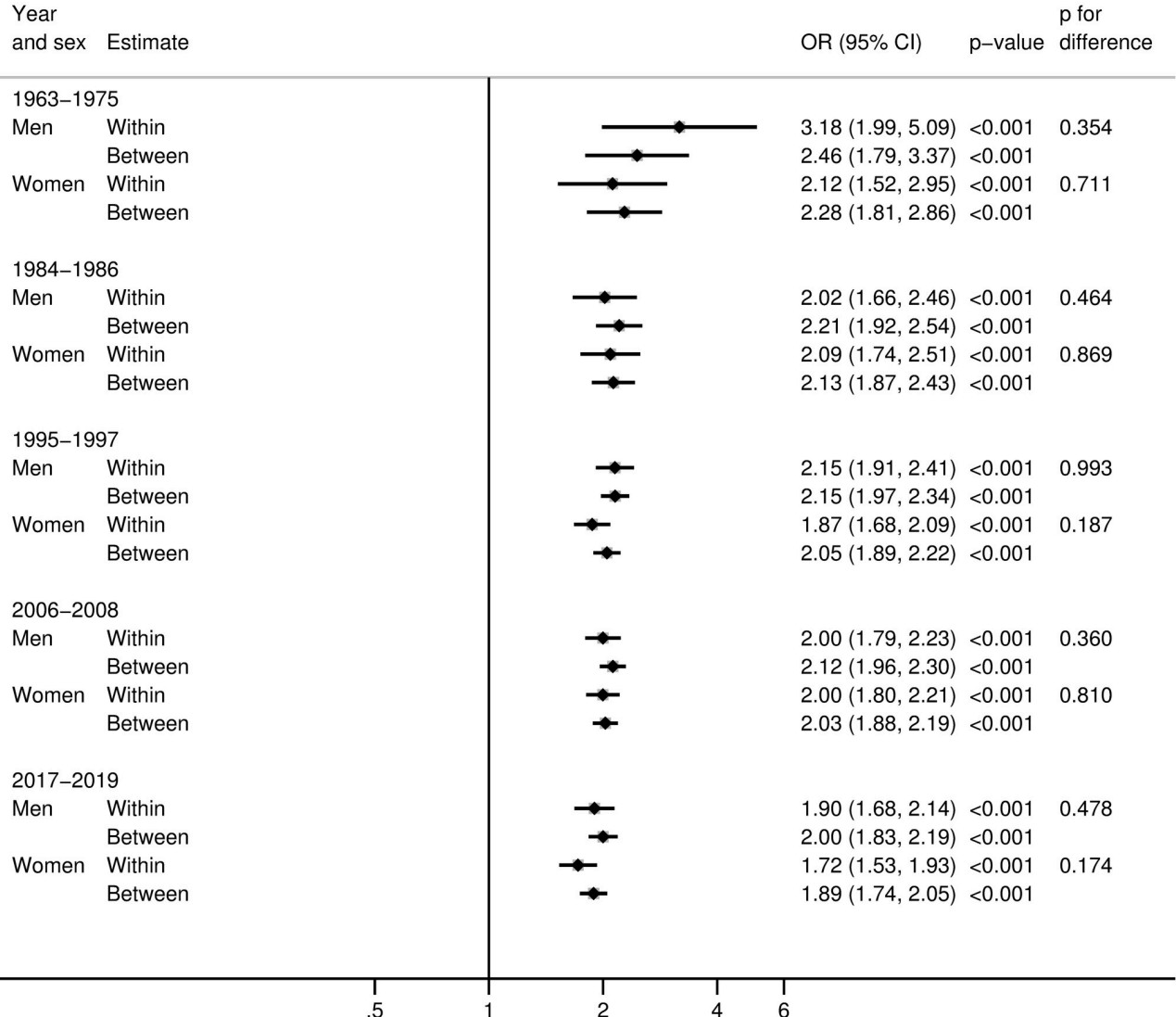

**Fig 2. Estimated OR (with 95% CI) for obesity per SD higher GPS for BMI within and between siblings by year.** Based on 29,585 individuals comprising 11,857 sibling groups within participants in the HUNT Study. BMI, body mass index; CI, confidence interval; GPS, genome-wide polygenic score; HUNT, Trøndelag Health Study; OR, odds ratio; SD, standard deviation.

## Discussion

In this study, we observed that from the 1960s to the late 2010s, the prevalence of obesity and severe obesity increased dramatically for the genetically predisposed yet remained relatively unchanged for the least genetically predisposed. For 35-year-old men and women, the absolute increase in prevalence of obesity using the GPS was 20 percentage points greater for the most genetically predisposed tenth compared with the least predisposed tenth. This suggests an increasing genetic inequality in obesity and severe obesity over time, consistent with the increasingly obesogenic environment [5] and the increasing variance in BMI seen over time in many countries [26]. Interestingly, the increase in severe obesity across GPS tenths was strongest among women. Conceptualizing the year of assessment as a broad indicator of the obesogenic environment, our study illustrates that despite being a very heritable trait, body weight

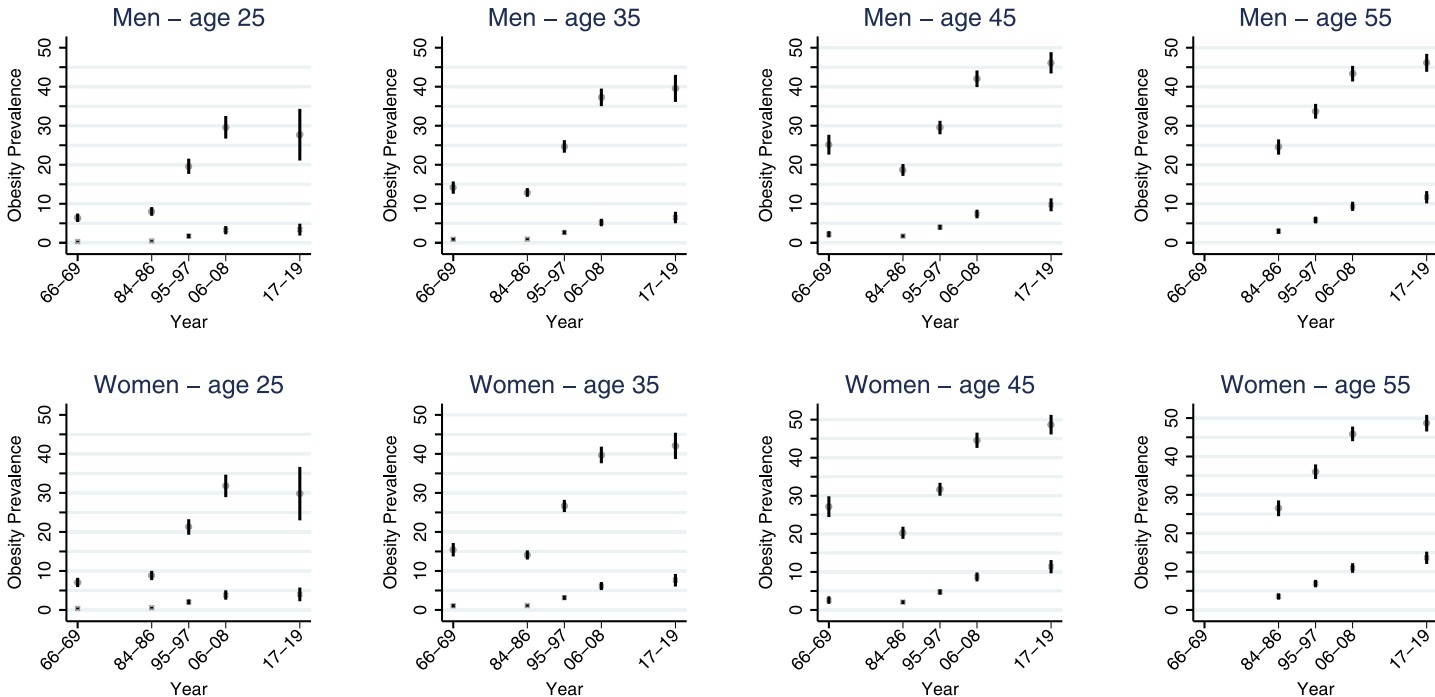

**Fig 3. Estimated prevalence of obesity by top and bottom tenths of GPS.** Estimated prevalence (%, with 95% CI) of obesity (BMI $\geq$ 30 kg/m$^2$) by top (most susceptible, circle) and bottom tenths (least susceptible, x) of GPS by age and time point for 31,717 men and 35,393 women who participated in the HUNT Study, Norway. *Youngest observed age in 2017–2019 was 28.6 years. BMI, body mass index; CI, confidence interval; GPS, genome-wide polygenic score; HUNT, Trøndelag Health Study.

seems modifiable by obesogenic exposure. Further, our findings demonstrate an interplay between genes and the environment that is robust to family-level confounding using sibling design.

## Comparison with other studies

In the Norwegian population of today, we found similarly explained variance for BMI and weight gradients across polygenic score tenths as previously described in contemporary British populations [10]. Although we acknowledge that both populations are of European decent, this affirms the GPS in a different cultural–geographic region. It is however notable that participation in the UK Biobank study is comparatively low (5%) and may be subject to participation bias where higher levels of adiposity reduced participation [27]. Compared to the British study, our study lacks statistical power in the younger age groups and could not replicate findings of an increasing weight gradient across polygenic score tenths from childhood to early adulthood [10]. Our dataset is however robust from age 25 to 75 years and does not affirm any clear age trends.

Combining the GPS with the dimension of time, our study suggests greater amplification of the effect of genetic predisposition on BMI during the obesity epidemic than previously shown [8,9]. Today's prevalence of obesity is a net result of the effects of the obesogenic environment added to genetic differences plus the interplay between genes and the environment. The relatively stable prevalence of obesity from the 1960s to the 1980s in our study could reflect a relatively stable environment with a constant genetic contribution to BMI in the population. From the mid-1980s, Norway experienced increased prosperity resulting in new working cultures, increased market consumption and automobile transport, as well as an unfavorable change in dietary patterns [28,29]. With the introduction of an obesogenic environment, the prevalence

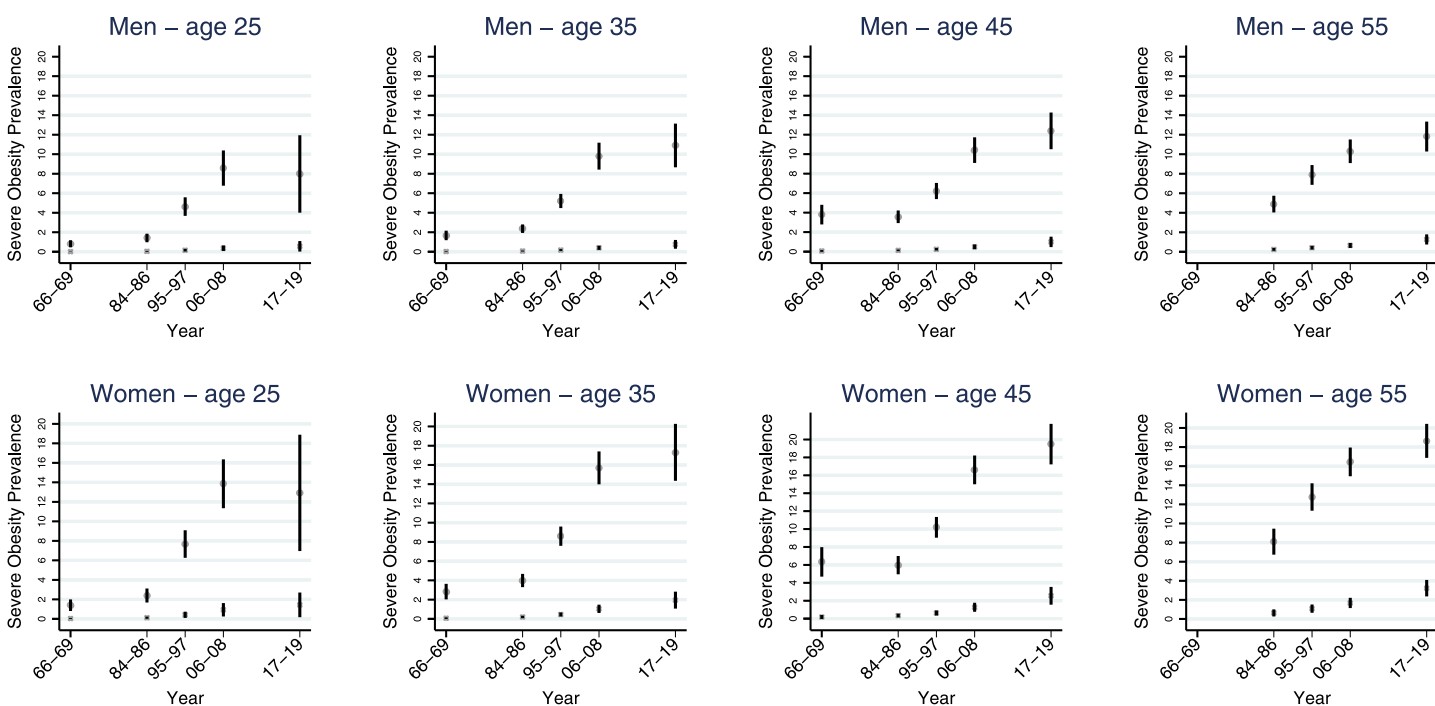

**Fig 4. Estimated prevalence of severe obesity by top and bottom tenths of GPS.** Estimated prevalence (%, with 95% CI) of severe obesity (BMI $\geq$ 35 kg/m$^2$) by top (most susceptible, circle) and bottom tenths (least susceptible, x) of GPS by age and time point for 31,717 men and 35,393 women who participated in the HUNT Study, Norway *Youngest observed age in 2017–2019 was 28.6 years. BMI, body mass index; CI, confidence interval; GPS, genome-wide polygenic score; HUNT, Trøndelag Health Study.

of obesity increased but so did the discrepancy in obesity status between the genetically predisposed and non-predisposed. These trends then level off as the prevalence of obesity stabilized over the last decade. In line with previous studies, we find a change in the distribution of BMI with an increasingly positive skew [26,30]. Our findings comply with a recent twin study collaboration suggesting unchanged heritability estimates for BMI over time and geography as a result of both increasing average BMI and an increasing impact of the environment on the effects of genetic variation [31,32]. A possible explanation comes from another study suggesting that the effect of certain genetic variants associated with obesity increase in people with higher BMI, and the enhanced genetic effects stem predominantly from gene by environment interactions [33].

## Strengths and limitations of this study

Our study provides statistical evidence for the changing effect of genetics on obesity over time. Although previous studies indicated that genetic variants known to predict BMI had larger effects after the onset of the obesity epidemic than before [8,9], to the best of our knowledge, this study includes the largest sample size and range of years of assessments and ages to date. Compared with our previous analysis of the HUNT Study population [9], BMI measurements as late as 2019 illustrate how genetic inequality in BMI stabilizes as the population weight levels off. Our study sample is considered unselected and is little affected by nonparticipation bias or bias from selective survival to date of genetic testing, as previously described [9]. For the eldest cohorts, we acknowledge a weak association between BMI measured in the 1960s and survival to and participation in genetic analyses in the 1990s. Still, we found little evidence of selection bias in analyses using genetic data from first-degree relatives as a proxy for those who did not

**Table 4. Percentage point increase in the prevalence of obesity and severe obesity over time for the tenths with the highest and lowest genetic susceptibilities.**

| Obesity | | | Men | | | | Women | | | |
|---|---|---|---|---|---|---|---|---|---|---|
| Time range | Age | GPS decile | Increase in prevalence | 95% CI | | p-value | Increase in prevalence | 95% CI | | p-value |
| 1966–2019 | 25 | 0 | 3.04 | 1.57 to 4.51 | | <0.001 | 3.58 | 1.90 to 5.27 | | <0.001 |
| | | 9 | 21.21 | 14.64 to 27.78 | | <0.001 | 22.62 | 15.81 to 29.42 | | <0.001 |
| | 35 | 0 | 5.55 | 4.06 to 7.03 | | <0.001 | 6.49 | 4.88 to 8.10 | | <0.001 |
| | | 9 | 25.29 | 21.61 to 28.97 | | <0.001 | 26.43 | 22.73 to 30.14 | | <0.001 |
| | 45 | 0 | 7.52 | 5.69 to 9.35 | | <0.001 | 8.73 | 6.79 to 10.66 | | <0.001 |
| | | 9 | 20.81 | 17.23 to 24.38 | | <0.001 | 21.36 | 17.74 to 24.99 | | <0.001 |
| 1984–2019 | 55 | 0 | 8.65 | 7.09 to 10.22 | | <0.001 | 9.99 | 8.37 to 11.61 | | <0.001 |
| | | 9 | 21.39 | 18.81 to 23.97 | | <0.001 | 21.97 | 19.36 to 24.58 | | <0.001 |
| | 65 | 0 | 7.52 | 5.92 to 9.13 | | <0.001 | 8.65 | 6.96 to 10.35 | | <0.001 |
| | | 9 | 12.72 | 9.71 to 15.73 | | <0.001 | 12.99 | 9.91 to 16.07 | | <0.001 |

| Severe obesity | | | Men | | | | Women | | | |
|---|---|---|---|---|---|---|---|---|---|---|
| Time range | Age | GPS decile | Increase in prevalence | 95% CI | | p-value | Increase in prevalence | 95% CI | | p-value |
| 1966–2019 | 25 | 0 | 0.54 | 0.01 to 1.07 | | 0.044 | 1.39 | 0.15 to 2.63 | | 0.028 |
| | | 9 | 7.12 | 3.18 to 11.06 | | <0.001 | 11.44 | 5.52 to 17.37 | | <0.001 |
| | 35 | 0 | 0.73 | 0.29 to 1.17 | | 0.001 | 1.87 | 1.00 to 2.74 | | <0.001 |
| | | 9 | 9.18 | 6.94 to 11.42 | | <0.001 | 14.38 | 11.32 to 17.43 | | <0.001 |
| | 45 | 0 | 0.93 | 0.41 to 1.44 | | <0.001 | 2.35 | 1.38 to 3.33 | | <0.001 |
| | | 9 | 8.52 | 6.49 to 10.56 | | <0.001 | 13.03 | 10.26 to 15.80 | | <0.001 |
| 1984–2019 | 55 | 0 | 1.02 | 0.47 to 1.56 | | <0.001 | 2.57 | 1.69 to 3.45 | | <0.001 |
| | | 9 | 6.85 | 5.32 to 8.37 | | <0.001 | 10.42 | 8.37 to 12.48 | | <0.001 |
| | 65 | 0 | 0.89 | 0.40 to 1.38 | | <0.001 | 2.24 | 1.32 to 3.16 | | <0.001 |
| | | 9 | 3.48 | 1.78 to 5.18 | | <0.001 | 5.25 | 2.75 to 7.74 | | <0.001 |

The percentage point increase is the difference between the estimated prevalence at the earliest time point subtracted from the estimated prevalence at the most recent time point.

CI, confidence interval; GPS, genome-wide polygenic score.

**Table 5. Difference in percentage point increase of prevalence in obesity and severe obesity between the tenths with the highest and lowest genetic susceptibilities over time.**

| Obesity | | Men | | | | Women | | | |
|---|---|---|---|---|---|---|---|---|---|
| Time range | Age | Difference in increase | 95% CI | | p-value | Difference in increase | 95% CI | | p-value |
| 1966–2019 | 25 | 18.24 | 12.43 to 24.05 | | <0.0001 | 19.11 | 13.19 to 25.03 | | <0.0001 |
| | 35 | 19.84 | 16.19 to 23.49 | | <0.0001 | 20.04 | 16.37 to 23.70 | | <0.0001 |
| | 45 | 13.40 | 9.47 to 17.33 | | <0.0001 | 12.74 | 8.73 to 16.76 | | <0.0001 |
| 1984–2019 | 55 | 8.96 | 5.90 to 12.02 | | <0.0001 | 8.19 | 5.06 to 11.32 | | <0.0001 |
| | 65 | 5.34 | 2.11 to 8.58 | | 0.0012 | 4.48 | 1.16 to 7.79 | | 0.0081 |

| Severe obesity | | Men | | | | Women | | | |
|---|---|---|---|---|---|---|---|---|---|
| Time range | Age | Difference in increase | 95% CI | | p-value | Difference in increase | 95% CI | | p-value |
| 1966–2019 | 25 | 6.60 | 2.85 to 10.36 | | 0.0006 | 10.10 | 4.62 to 15.57 | | 0.0003 |
| | 35 | 8.49 | 6.27 to 10.71 | | <0.0001 | 12.57 | 9.57 to 15.58 | | <0.0001 |
| | 45 | 7.66 | 5.58 to 9.74 | | <0.0001 | 10.76 | 7.88 to 13.63 | | <0.0001 |
| 1984–2019 | 55 | 4.00 | 2.40 to 5.59 | | <0.0001 | 5.18 | 2.90 to 7.48 | | <0.0001 |
| | 65 | 2.67 | 0.94 to 4.39 | | 0.0024 | 3.11 | 0.57 to 5.65 | | 0.0163 |

For each age category and each sex, the estimated difference in percentage point increase equals the difference between percentage points increase in prevalence over time for the highest compared to the lowest genetically predisposed tenth presented in Table 4.

CI, confidence interval.

participate in genetic testing. Also, results for 25-year-old men and women in 2017 to 2019 are extrapolated from a broader age range.

Combining this unique dataset with such a powerful polygenic predictor is the principal strength of our study. Although the GPS does not account for the effect of rare gene variants recently recovered by whole-genome sequencing [6], it is the first genetic instrument to provide meaningful predictive power by encompassing over 2 million common gene variants associated with obesity. Compared with the previous score [9], the increase in explained variance from 3% to 9% may appear small. However, this reflects a substantially greater difference in BMI between the genetically predisposed and lesser predisposed and improves the classification of genetic risk considerably (Tables E and F in S2 Text). Among today's middle-aged adults, this difference amounts to a 13-kg weight gradient across polygenic score tenths [10]. We applied the contemporary GPS to past time periods in the absence of historical data from a separate population. Ideally, with access to a GPS from the past, we could examine if the increased effect of genetic risk on BMI still occurs. Future research should focus on constructing a polygenic risk score from historical data.

In this study, we validate our findings in within-sibship analyses, which suggest that neither assortative mating, dynastic effects, nor fine-scale population stratification issues generate the results we see [14,15,34]. However, we recognize that parents of participants in our cohort met before the start of the obesity epidemic in the mid-1980s, and genotypic assortment for BMI might be a greater issue in the future.

## Generalizability of the findings

The effect of the obesogenic environment on population weight acts partly through enhancing the effect of our genes [9]. The magnitude seems to relate to the obesogenic exposure in the macroenvironment, conceptualized as year of assessment in this study. In contrast to most countries [5,35], the prevalence of obesity in Norway stabilized over the last decade. As a result, we observe a stabilizing gradient in weight and only a slight increased risk of obesity across polygenic score tenths. Although difficult to measure, this finding could indicate a consistent obesogenic exposure in the macroenvironment in recent years. However, in countries where obesity prevalence is still increasing, these weight gradient trends across genetic susceptibility would likely be different.

## Implications and future research, clinical practice, and public policy

Our findings suggest a genetic inequality in obesity during the obesity epidemic in Norway. This novel insight may help identify those susceptible to environmental change as a preventative measure.

Future research should focus on specific gene environment interactions that could help determine which preventative efforts are most effective. Regardless, secular trends have increased body weight for both genetically predisposed and genetically non-predisposed people. It is time for the global community to recognize and to address the determinants of ill health that foster the unhealthy environment in which we live [5,36].

## Conclusions

The prevalence of obesity increased substantially from the mid-1980s and has stabilized to a new level over the last decade in the Norwegian population. While obesity is a highly heritable trait [6], our study illustrates how it is still modifiable by the obesogenic exposure. Utilizing the substantial changes in our environment over time, we expose a growing inequality in risk for obesity and severe obesity between the genetically predisposed and lesser predisposed. The

magnitude of our findings using the GPS is far greater than previously anticipated, holds true over time, and is robust to confounding.

## Supporting information

**S1 Protocol. Prospective protocol.**
(DOCX)

**S1 Text. Modifications to the prospective protocol with rationale.**
(DOCX)

**S2 Text. Supporting figures and tables.**
(DOCX)

**S1 Checklist. Strengthening the reporting of observational studies in epidemiology.**
(DOC)

## Acknowledgments

The Trøndelag Health Study (the HUNT Study) is a collaboration between HUNT Research Centre, (Faculty of Medicine and Health Sciences, NTNU, Norwegian University of Science and Technology), the Nord-Trøndelag County Council, the Central Norway Regional Health Authority, and the Norwegian Institute of Public Health. The genotyping in HUNT was financed by the National Institutes of Health (NIH); University of Michigan; The Research Council of Norway; the Liaison Committee for Education, Research and Innovation in Central Norway; and the Joint Research Committee between St. Olavs University Hospital and the Faculty of Medicine and Health Sciences, NTNU. The genotype quality control and imputation has been conducted by the K.G. Jebsen Center for Genetic Epidemiology, Department of Public Health and Nursing, Faculty of Medicine and Health Sciences, Norwegian University of Science and Technology (NTNU).

## Author Contributions

**Conceptualization:** Johan Håkon Bjørngaard, George Davey Smith, Bjørn Olav Åsvold, Erik R. Sund, Gunnhild Åberge Vie.

**Data curation:** Cristen J. Willer.

**Formal analysis:** Maria Brandkvist, Ben Brumpton, Gunnhild Åberge Vie.

**Funding acquisition:** Johan Håkon Bjørngaard.

**Investigation:** Maria Brandkvist, Gunnhild Åberge Vie.

**Methodology:** Maria Brandkvist, Johan Håkon Bjørngaard, Ben Brumpton, George Davey Smith, Bjørn Olav Åsvold, Gunnhild Åberge Vie.

**Project administration:** Maria Brandkvist, Gunnhild Åberge Vie.

**Supervision:** Johan Håkon Bjørngaard, Rønnaug Astri Ødegård, Gunnhild Åberge Vie.

**Validation:** Maria Brandkvist, Johan Håkon Bjørngaard, Rønnaug Astri Ødegård, Ben Brumpton, George Davey Smith, Bjørn Olav Åsvold, Erik R. Sund, Kirsti Kvaløy, Cristen J. Willer, Gunnhild Åberge Vie.

**Visualization:** Maria Brandkvist, Gunnhild Åberge Vie.

**Writing – original draft:** Maria Brandkvist.

**Writing – review & editing:** Maria Brandkvist, Johan Håkon Bjørngaard, Rønnaug Astri
Ødegård, Ben Brumpton, George Davey Smith, Bjørn Olav Åsvold, Erik R. Sund, Kirsti
Kvaløy, Cristen J. Willer, Gunnhild Åberge Vie.

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
