## [Editor Report · Decision Letter 0]

21 May 2020

Dear Dr Brandkvist, 

Thank you for submitting your manuscript entitled "Genetic inequalities in obesity and severe obesity during the obesity epidemic: longitudinal findings from the HUNT Study verified by sibling design" for consideration by PLOS Medicine.

Your manuscript has now been evaluated by the PLOS Medicine editorial staff, as well as by an academic editor with relevant expertise, and I am writing to let you know that we would like to send your submission out for external peer review.

Feel free to email us at plosmedicine@plos.org, or 'reply all' to this message if you have any queries relating to your submission.

Kind regards,

Caitlin Moyer, Ph.D.,

Associate Editor

PLOS Medicine

---

## [Decision Letter · Decision Letter 1]

7 Jul 2020

Dear Dr. Brandkvist,

Thank you very much for submitting your manuscript "Genetic inequalities in obesity and severe obesity during the obesity epidemic: longitudinal findings from the HUNT Study verified by sibling design" (PMEDICINE-D-20-01911R1) for consideration at PLOS Medicine. 

Your paper was evaluated by a senior editor and discussed among all the editors here. It was also discussed with an academic editor with relevant expertise, and sent to three independent reviewers, including a statistical reviewer. The reviews are appended at the bottom of this email and any accompanying reviewer attachments can be seen via the link below:

[LINK]

In light of these reviews, I am afraid that we will not be able to accept the manuscript for publication in the journal in its current form, but we would like to consider a revised version that addresses the reviewers' and editors' comments. Obviously we cannot make any decision about publication until we have seen the revised manuscript and your response, and we plan to seek re-review by one or more of the reviewers. 

We expect to receive your revised manuscript by Jul 28 2020 11:59PM. Please email us (plosmedicine@plos.org) if you have any questions or concerns.

We look forward to receiving your revised manuscript. 

Sincerely,

Emma Veitch, PhD

PLOS Medicine

On behalf of Clare Stone, PhD, Acting Chief Editor,

PLOS Medicine

plosmedicine.org

*One reviewer comments on the overlap between the analyses in this paper and those in a recent BMJ article, focussing on the effect of a 97-variant genetic risk score and obesity risk. The editors felt that these concerns did not invalidate the paper from further consideration in PLOS Medicine but it would be good to try to be more clear in the paper what is the value and novel contribution of the present genome-wide score. 

*Please include a brief note in the last sentence of the Abstract Methods and Findings section summarising any key limitation(s) of the study's methodology.

*At this stage, we ask that you include a short, non-technical Author Summary of your research to make findings accessible to a wide audience that includes both scientists and non-scientists. The Author Summary should immediately follow the Abstract in your revised manuscript. This text is subject to editorial change and should be distinct from the scientific abstract. Please see our author guidelines for more information: https://journals.plos.org/plosmedicine/s/revising-your-manuscript#loc-author-summary

*Please clarify, ideally in the Methods section, if the analytical approach used in this paper corresponded to one laid out in a prospective protocol or analysis plan? Please state this (either way) early in the Methods section.

Comments from the reviewers:

Reviewer #1: Reviewer comments provided in an attached document.

Reviewer #2: Brandkvist et al. demonstrate an increase in effect of a genetic score for BMI over time. The study is well written and the methods appear adequate for the research question. I only have minor comments that I feel should be adressed before the manuscript is accepted for publication.

The main result, an increase in genetic effect over time, is interpreted as a consequence of an increasingly obesogenic environment. I agree that this is the likely causal factor. However the authors need to argue from reported environmental observations that are relevant for the cohort used in the study. I would appreciate references to reports that have measured changing habits in activity and food intake for Norwegians, preferably in the Nord-Trøndelag region or a comparable population such as other Scandinavians, for the time period in question. 

In addition, the phrasing in the discussion: "Our findings show an interplay between genes and the environment [..]", should be modified since environmental factors were not included in statistical models. Rather, they were inferred by assumption of changes in environment over time with no reference to measured or reported data on environmental factors. 

Figures 3 and 4 are missing legends.

Reviewer #3: Study by Brandkvist and colleagues:

1-The authors recently published a study investigating the same question with a 97 SNP-based polygenic score versus a genome-wide polygenic score here. As these scores respectively explain 2.7% and 9% of BMI variation in the population, we do not expect very different conclusions from the two studies. In that context, the results of the current study, while interesting, have a limited incremental value. 

2-The authors concluded that the increase in the prevalence of obesity during the 60 years follow-up was steeper among the genetically predisposed (low genetic risk: 35 year-old men: from 1% to 7%, 35 year-old women: from 1% to 8%; high genetic risk: 35 year-old men: from 14% to 40%; 35 year-old women: from 15% to 42%). This statement is true if we consider absolute changes in the prevalence of obesity. However, another way to look at the data is to consider that the prevalence of obesity was multiplied by 7-8 fold in the low genetic risk group versus 2.8-2.85 fold in the high genetic risk group during the last decades. With these data in mind, it looks like the obesogenic environment had a more detrimental impact on the development of obesity in the low genetic risk than the high genetic risk group ('environmental' obesity). Interpreting the data while considering the relative versus absolute changes in the prevalence of obesity is probably worth it. 

3-The authors focused on 35-year old populations as it maximizes the statistical power. It may be relevant to perform sensibility analyses to make sure that the results observed in this age group can be generalized to younger and older people in the HUNT study.

4-'adults…children…are overweight or obese'. Please use people first language (adults…children…are having overweight or obesity').

5-The authors mention a heritability estimate of 40% for BMI. Heritability can be measured using different designs (twins, families, general populations). Please mention a heritability range of 40-75% (see the recent meta-analysis of heritability studies by Strijecky et al. Obes Rev 2018).

6-'fat mass, as indicated by body mass index'. BMI is a poor surrogate of fat mass in non-obese populations. I suggest rephrasing the sentence. The authors can focus on BMI heritability estimates, or can cite heritability studies for fat mass.

7-As BMI equals weight divided by height square, it is questionable to adjust BMI for height in my opinion (lack of independence between the covariate and the outcome).

8-The authors did not mention any transformation for BMI (ex. logarithmic, ranked-based inverse normal transformation). These transformations are frequently used in literature to correct for the lack of normality of the BMI distribution. 

9-The authors may describe in more detail what were the changes of weight / BMI across time both globally and in the genetic subgroups in the Results section.

10-In the discussion, the authors may cite and comment the paper by Abadi et al. Am J Hum Genet 2017. This study suggests the existence of 'snowball' obesity genes that may provide a nice mechanistic explanation for the results of the HUNT study.

[LINK]

---

## [Decision Letter · Decision Letter 2]

14 Sep 2020

Dear Dr. Brandkvist,

Thank you very much for submitting your manuscript "Genetic inequalities in obesity and severe obesity during the obesity epidemic: longitudinal findings from the HUNT Study verified by sibling design" (PMEDICINE-D-20-01911R2) for consideration at PLOS Medicine. 

Your paper was evaluated by a senior editor and discussed among all the editors here. It was also discussed with an academic editor with relevant expertise, and sent to a statistical reviewer for re-review. The reviews are appended at the bottom of this email and any accompanying reviewer attachments can be seen via the link below:

[LINK]

Although we note that Reviewer 1 has no further concerns, a few remaining issues to address were noted by the Academic Editor and editorial team. I am afraid that we will not be able to accept the manuscript for publication in the journal in its current form, but we would like to consider a revised version that fully addresses the editors' comments. Obviously we cannot make any decision about publication until we have seen the revised manuscript and your response. 

In revising the manuscript for further consideration, your revisions should fully address the specific points made by the editors. Please also check the guidelines for revised papers at http://journals.plos.org/plosmedicine/s/revising-your-manuscript for any that apply to your paper. In your rebuttal letter you should indicate your response to the editors' comments, the changes you have made in the manuscript, and include either an excerpt of the revised text or the location (eg: page and line number) where each change can be found. Please submit a clean version of the paper as the main article file; a version with changes marked should be uploaded as a marked up manuscript.

We expect to receive your revised manuscript by Sep 28 2020 11:59PM. Please email us (plosmedicine@plos.org) if you have any questions or concerns.

We look forward to receiving your revised manuscript. 

Sincerely,

Caitlin Moyer, PhD

Associate Editor 

PLOS Medicine

plosmedicine.org

1.Comments from the Academic Editor: Response to Reviewer 3: Please include tables 1 and 2 from your response to Reviewer 3’s comments in the text of the manuscript (comparing the performance of the earlier GRS versus the more recent polygenic risk score) as this is helpful to emphasizing the advance of this study compared to previous reports using genetic risk score.

2.Comments from the Academic Editor: Please make it clear in the text why a number of significantly younger individuals with lower BMI from YHUNT2 were included in a dataset of otherwise older individuals (see table S1 and S3).

3. Response to Reviewer 3: Please add the additional sensitivity analysis done with BMI not adjusted for height to the supporting information files and refer to these in the Methods and Results (in response to Reviewer 3 comment 7).

4. Competing Interests: Please add this statement to the manuscript's Competing Interests: "GDS is an Academic Editor on PLOS Medicine's editorial board.”

5. Abstract: Methods and Findings: In the first sentence, a comma is missing from “67 110 individuals” (The comma is similarly missing from this number in the Author Summary, Methods, and Results sections).

6. Abstract: Methods and Findings: Please revise this sentence, as your study did not formally test a gene x environment interaction. We suggest: “The greater increase in BMI in genetically predisposed individuals over time was apparent after adjustment for family-level confounding using a sibling design.” or similar.

7. Abstract: Methods and Findings: Please clarify what is meant by “slight differential survival to date of genetic testing” in the limitations statement.

8. Abstract: Methods and Findings: Please include both 95% CIs and p values for all results presented here.

9. Abstract: Conclusions: * Please address the study implications without overreaching what can be concluded from the data; For the Conclusions paragraph, we suggest the following revision; “In the context of increasingly obesogenic changes in our environment over six decades, our findings reveal a growing inequality in risk for obesity and severe obesity across genome-wide polygenic score tenths. Our results suggest that while obesity is a heritable trait at least in part, it is still modifiable by environmental factors. Whilst it may be possible to identify those most susceptible to environmental change, who thus have the most to gain from preventive measures, efforts to reverse the obesogenic environment will benefit the whole population and help resolve the obesity epidemic.”

10. Author Summary: Please structure the Author Summary using bullet points for each point, and restrict the number of points to 3 per section, and 1 sentence per bullet point, if possible. Please see our author guidelines for more information: https://journals.plos.org/plosmedicine/s/revising-your-manuscript#loc-author-summary

11. Author Summary: Why was this study done? In the 3rd point of this section, please remove the word "powerful" " In the 4th point of this section, please remove “vastly more powerful” as this is not informative.

12.Author Summary: What did the researchers do and find?: Please revise to (or similar):

--We assessed the relationship between an obesogenic environment and the prevalence of obesity according to genetic predisposition over six decades in Norway, with increasing and stabilizing prevalence of obesity, using a genome-wide polygenic score for BMI and validating by sibling design.

--Using genetic data from 67,110 individuals aged 13 to 80 years with repeated height and weight measurements recorded between 1966 and 2019, we found that the prevalence of obesity differed between the participants with the highest and lowest genetic susceptibility to obesity for all ages at each decade, and the difference increased gradually from the 1960s to the 2000s to then stabilize over the last decade

.

--For example, for 35 year old men and women, the increase in prevalence of obesity was 20 percentage points greater for the most genetically predisposed tenth compared with the least predisposed tenth.

13. Citations: For in-text citations, please use square brackets rather than parentheses and place citations before the punctuation, like this [1].

14. Introduction: First paragraph: Please replace the terms “developing” and “western world” by referring to high income countries rather than "developed" or "Western" countries, and to low or middle income countries rather than "developing countries"

15. Introduction: Last paragraph: Please revise to “This is important as it mitigates bias…”

16. Introduction: Please conclude the Introduction with a clear description of the study question or hypothesis. Final sentence: Please revise to remove “the most powerful” as this is hyperbole. We suggest: “Hence, the goal of this study is to quantify and validate the interplay between our genes and the environment.” or similar.

17.Methods: First paragraph: Please clarify the ages of the participants, as the first sentence states you included individuals aged 13 or older, and the last sentence indicates you did not include individuals aged younger than 14. (Please also be sure this is consistent throughout the manuscript)

18. Methods: Ethical approval: Please note the nature of informed consent (e.g. written informed consent).

19. Methods: Page 10: Please provide additional details on how the sibling analyses were done and how this strengthens the study.

20. Methods: Statistical analysis: Please specify the significance level used to determine whether there were differences in your results (eg, P<0.05, two-sided) as well as the statistical test used to derive a p value.

21. Results: Please present the findings quantified with 95% CIs and p values to reflect statistical differences of comparisons referred to in the text. Specifically, please provide quantitative results, 95% CIs and p values for the increase in BMI/prevalence of obesity over time overall and for genetically predisposed vs. not predisposed groups. Please provide both the height adjusted and unadjusted analyses.

22. Results: Please present the results of sibling analyses with 95%CIs and p values “We found comparable associations between polygenic risk score and BMI as well as obesity within and between sibling groups with little evidence of bias from assortative mating, population stratification or dynastic effects (Fig 2, S6 Fig)”

23. Results: Page 12: Please provide the statistical results to substantiate this sentence, or change this to “observed to be steeper” or “apparently steeper” to indicate that this was not statistically tested. “The increase in prevalence of obesity and severe obesity was steeper among the genetically predisposed over the time period (Fig 3, Fig 4).”

24. Results Page 12-13: Please provide p values in addition to 95% CIs for absolute changes in obesity prevalence.

25. Results: Page 12: Please remove the terms “dramatically” and “substantially” from the following sentences. If statistical significance is meant, please replace with “significantly” where appropriate. “From relative stability in the 1960s to 1980s, the weight for both the genetically predisposed and nonpredisposed increased dramatically from the mid-1980s to the 2000s and then stabilized to a higher level over the past decade. Height-adjusted BMI differed substantially across polygenic score tenths for all ages and at each decade, and the difference varied proportional to the changes in population weight (S5 Fig, S2 Table).”

26. Discussion: Please present and organize the Discussion as follows: a short, clear summary of the article's findings; what the study adds to existing research and where and why the results may differ from previous research; strengths and limitations of the study; implications and next steps for research, clinical practice, and/or public policy; one-paragraph conclusion.

27. Discussion: Please revise the first sentence to: In this study, we observed that, in transitioning to an obesogenic environment from the 1960s to the late-2010s, the prevalence of obesity and severe obesity increased dramatically for the genetically predisposed yet remained relatively unchanged for the least genetically predisposed.”

28.Discussion: Page 14: Please remove “the most powerful” from this sentence as it is hyperbole. “Combining this unique dataset with the most powerful polygenic predictor to date is the principal strength of our study.” Please also remove other instances of hyperbole.

29. Discussion: please either remove or clarify this sentence, as the meaning is unclear in the context of your results: “Despite being a very heritable trait, our study illustrates that body weight is modifiable proportionate to the presumed degree of the obesogenic exposure.”

30. Discussion: please either remove or clarify these sentences, as it isn’t clear that your results support this conclusion: “The effect of the obesogenic environment on population weight acts partly through enhancing the effect of our genes. The magnitude seems to relate directly with degree of obesogenic exposure in the macroenvironment.”

31. References: Please use the "Vancouver" style for reference formatting, and see our website for other reference guidelines https://journals.plos.org/plosmedicine/s/submission-guidelines#loc-references

In particular, please check the formatting of ref. #5 for missing information, and please provide English translations for 27 and 28.

32. Checklist: Thank you for including the completed STROBE checklist as Supporting Information. Please revise the checklist, please use section and paragraph numbers (e.g. “Methods, paragraph 1”), rather than page numbers to refer to locations in the text.

Please add the following statement, or similar, to the Methods: "This study is reported as per the Strengthening the Reporting of Observational Studies in Epidemiology (STROBE) guideline (S1 Checklist)."

33. Figure 2: Please provide p values to accompany the ORs reported. Please note in the legend what is represented by each of the 5 panels.

34. Figures 3 and 4: Please provide X axis labels. Please revise the Y axis label to read “Obesity/Severe Obesity Prevalance” or similar. Please use (and explain in the figure legend) different markers or colors to indicate the points and 95% CIs corresponding to the highest and lowest propensity scores.

35. S2 Figure: Please provide Y axis labels.

36. S3 Figure: Is there a chance some of the lines are out of order? The trend of increasing BMI over time doesn’t fit with this plot of BMI distribution for the entire study sample by time of measurement, in particular the 2000-2001 line seems out of place (if this is due to the fact that this was the year when only adolescents participated, please make note of this in the legend).

37. S5 Figure: Please use different colors or different marks (and explain in the legend) to indicate the top vs. bottom score data. In the legend, please revise 28,6 to 28.6.

38. S6 Figure: Please indicate in the legend the purpose of the different panels, and please include p values along with the 95% CIs. Please clarify if the X axis is “height adjusted BMI”

39. S7, S8, and S9 Figures: Please use different colors or different marks (and explain in the legend) to indicate the top vs. bottom score data. Please provide X axis labels for all plots.

40. Supporting Information Files: The supplemental information files S1 Table and S2 Table. S3 Table, S4 Table are central to the understanding of the paper. Please incorporate these into the main paper. 

41. S1 Table: Please define abbreviations for SD and BMI in the legend.

42. S2 Table: Please also provide the p values associated with these differences. Please clarify if “BMI” represents the difference value, or another measure and provide units if applicable.

43. S3 Table: Please provide p values for these differences. Please clarify what is presented in the columns “Men” and “Women”

44. S4 Table: Please provide p values for these differences. Please make it more clear what is presented in the columns “Men” and “Women”

45. S5 Table: Please provide p values for these differences. Please indicate what is presented in column “BMI” and how this differs from the presentation of data in Table S2.

Comments from the reviewers:

Reviewer #1: I am satisfied with the manuscript and the edits the authors have made to address my previous comments. I therefore recommend that their work is accepted for publication.

[LINK]

---

## [Editor Report · Decision Letter 3]

13 Oct 2020

Dear Dr. Brandkvist,

Thank you very much for re-submitting your manuscript "Genetic inequalities in obesity and severe obesity during the obesity epidemic: longitudinal findings from the HUNT Study verified by sibling design" (PMEDICINE-D-20-01911R3) for review by PLOS Medicine.

I have discussed the paper with my colleagues and the academic editor. I am pleased to say that provided the remaining editorial and production issues are dealt with we are planning to accept the paper for publication in the journal.

[LINK]

In revising the manuscript for further consideration here, please ensure you address the specific points made by the editors. In your rebuttal letter you should indicate your response to the editors' comments and the changes you have made in the manuscript. Please submit a clean version of the paper as the main article file. A version with changes marked must also be uploaded as a marked up manuscript file.

We look forward to receiving the revised manuscript by Oct 20 2020 11:59PM. 

Sincerely,

Caitlin Moyer, Ph.D.

Associate Editor 

PLOS Medicine

plosmedicine.org

Requests from Editors:

1.From the academic editor: Thank you for including tables 1 and 2. These tables are important to include, but it seems they may be better as supporting information tables, rather than in the main manuscript.

2.From the academic editor: Please clarify further the response regarding the YHUNT2 young adults, suggesting that they were not included in some of these analyses. Your response in the first paragraph of the results is unclear: ”Hoping to fully utilize the wide age range of the dataset, we included BMI measurements of adolescents available from the 1960s and 2000-01. In the results however, we chose to focus on BMI measurements from adult participants as observations in the adolescent age range were limited (Table 3).” Please clarify whether (and if so, which of) the results include adolescent data or not

3.Title: Please revise to: "Genetic associations with temporal shifts in obesity and severe obesity during the obesity epidemic in Norway: a longitudinal population-based cohort (The HUNT study)"

4.Competing interests: Please remove the sentence: “All authors have completed the ICMJE uniform disclosure form at www.icmje.org/coi_disclosure.pdf.” And please change the final sentence from “All authors have no financial relationships with any organizations that might have an interest in the submitted work in the previous five years” to “The authors declare no other competing interests."

5.Data availability statement: Please revise the data availability statement to read: “"Data from the HUNT Study can be made available to qualified researchers upon request to the HUNT Data Access Committee (hunt@medisin.ntnu.no). The data availability policy for the HUNT study is available here: (http://www.ntnu.edu/hunt/data). Data are only available to research groups with a PI affiliated with a Norwegian research institute. The Norwegian Institute of Public Health will consider applications for data from the Tuberculosis screening program through www.helsedata.no. Linkages require ethical clearance by the Regional Ethical Committees, more information is available at https://rekportalen.no/."

6.Abstract: Background: Please revise to: “Our study investigates how changes in population weight and obesity over time are associated with genetic predisposition in the context of an obesogenic environment over six decades, and examines the robustness of the findings using sibling design.

7.Abstract: Please define abbreviations for BMI and HUNT at first use.

8.Abstract: Conclusions: In the second sentence, please remove the word “clearly”

9.Author summary: Please format this section using bullets for each point.

10.Author summary: What did the researchers do and find?: Please revise to” We assessed the changes in prevalence of obesity according to genetic predisposition over six decades…” as your study does not directly investigate the relationship with an obesogenic environment (though that can be an interpretation)

11.Author summary: What do these findings mean?: Please revise the first and second point to a single point: “The results indicate that over the past 6 decades, the least genetically predisposed people seem relatively protected from obesity and almost completely protected from severe obesity whereas the most predisposed people are at risk for both obesity and severe obesity, suggesting an interplay between genes and an increasingly obesogenic environment could play a role in growing differences in obesity risk between individuals with varying genetic predisposition.”

12.Author summary: What do these findings mean? Please revise the third point to: “The findings from this study highlights the need to identify and to address the specific factors that led to the population wide increase in obesity.”

13.In text citations: Please include a space between the word and the bracket of the reference, like this [1].

14. Introduction: 3rd paragraph: Please remove the word “vastly”

15. Methods: Patient and Public Involvement: Please clarify what is meant by “We will seek involvement from a patient organization in the development of an appropriate method of

Dissemination.”

16.Results: First sentence: There is a comma missing from “202 030 BMI measurements”

17. Results: Page 14- please clarify whether “weight” or “BMI” is intended here, as only BMI is presented in the table: “From relative stability in the 1960s to 1980s, the weight for both the genetically predisposed and nonpredisposed increased dramatically from the mid-1980s to the 2000s…”

18. Discussion: first paragraph: Please revise the following sentences in the first paragraph, because your study does not directly assess how the passing of time equates to an obesogenic environment- please revise to “In this study, we observed that, from the 1960s to the late-2010s, the prevalence of obesity and severe obesity increased dramatically for the genetically predisposed yet remained relatively unchanged for the least genetically predisposed.”

And “This suggests an increasing genetic inequality in obesity and severe obesity over time, consistent with the increasingly obesogenic environment[reference] and the increasing variance in BMI seen over time in many countries[26].”

19.Discussion: Page 22: Please qualify with “to the best of our knowledge” or similar: “...this study includes the largest sample size and range of years of assessments and ages to date…”

20.Discussion: Page 23: Please revise to: “In this study we validate our findings in within-sibship analyses, which suggest that neither assortative mating, dynastic effects, nor fine-scale population stratification issues generate the results we see[14, 15, 34].”

21.Figure 1: Please revise the bottom-most box of the figure. It is misleading as the study did not formally assess a gene x environment interaction analysis.

22.Figure 2: Please provide the ORs with 95% CIs on the right to accompany the p values.

23.References: Please use the "Vancouver" style for reference formatting, and see our website for other reference guidelines, particularly for for the formatting of bioRxiv postings, for example.

https://journals.plos.org/plosmedicine/s/submission-guidelines#loc-references

24.Table 1 and Table 2: Please define abbreviations for GPS and GRS in the legends. Please include legends that describe what is illustrated in the tables.

25.Table 3: Please provide additional demographic information on the study participants, such as comorbidities, lifestyle, and economic status measures.

26.Table 6: Please edit the typos in the 95% CI column for men (“toTo”), and please use consistent capitalization of “to” throughout.

[LINK]

---

## [Editor Report · Decision Letter 4]

5 Nov 2020

Dear Dr. Brandkvist, 

On behalf of my colleagues and the academic editor, Dr. Ronald C. Ma, I am delighted to inform you that your manuscript entitled "Genetic associations with temporal shifts in obesity and severe obesity during the obesity epidemic in Norway: a longitudinal population-based cohort (The HUNT study)" (PMEDICINE-D-20-01911R4) has been accepted for publication in PLOS Medicine. 

PRODUCTION PROCESS

Before publication you will see the copyedited word document (within 5 business days) and a PDF proof shortly after that. The copyeditor will be in touch shortly before sending you the copyedited Word document. We will make some revisions at copyediting stage to conform to our general style, and for clarification. When you receive this version you should check and revise it very carefully, including figures, tables, references, and supporting information, because corrections at the next stage (proofs) will be strictly limited to (1) errors in author names or affiliations, (2) errors of scientific fact that would cause misunderstandings to readers, and (3) printer's (introduced) errors. Please return the copyedited file within 2 business days in order to ensure timely delivery of the PDF proof. 

If you are likely to be away when either this document or the proof is sent, please ensure we have contact information of a second person, as we will need you to respond quickly at each point. Given the disruptions resulting from the ongoing COVID-19 pandemic, there may be delays in the production process. We apologise in advance for any inconvenience caused and will do our best to minimize impact as far as possible.

PRESS

PROFILE INFORMATION

Thank you again for submitting the manuscript to PLOS Medicine. We look forward to publishing it. 

Best wishes, 

Caitlin Moyer, Ph.D.

Associate Editor 

PLOS Medicine

plosmedicine.org